# Ultrasound Examination of Unilateral Seminoma in a Salernitano Stallion

**DOI:** 10.3390/ani12070936

**Published:** 2022-04-06

**Authors:** Brunella Anna Giangaspero, Roberta Bucci, Francesca Del Signore, Massimo Vignoli, Jasmine Hattab, Gina Rosaria Quaglione, Lucio Petrizzi, Augusto Carluccio

**Affiliations:** 1Faculty of Veterinary Medicine, University of Teramo, Piano d’Accio, 64100 Teramo, Italy; bagiangaspero@unite.it (B.A.G.); rbucci@unite.it (R.B.); jhattab@unite.it (J.H.); lpetrizzi@unite.it (L.P.); acarluccio@unite.it (A.C.); 2Operative Unite of Pathological Anatomy, ASL Teramo, 64100 Teramo, Italy; gina.quaglione@aslteramo.it

**Keywords:** stallion, testis, ultrasound, histopathology, seminoma

## Abstract

**Simple Summary:**

An 18-year-old Salernitano stallion was presented for enlargement of the left testicle with no other clinical signs. Sonoelastographic examination showed parenchymal changes with deformation of the normal testis, appearing with an irregular profile. Unilateral orchiectomy was performed. Considering gross, microscopic and immunohistochemical findings, a definitive diagnosis of diffuse seminoma was made. Three months later, follow-up assessment showed no evidence of recurrence, preserved reproductive abilities or fertility, but did show reduced testosterone levels.

**Abstract:**

An 18-year-old Salernitano stallion developed a progressive enlargement of the left testicle over eight months. An ultrasound evaluation was performed, along with a hormonal profile. A histopathological evaluation of the testis was performed after unilateral orchiectomy. On B-mode ultrasound examination, testicular parenchyma was characterized by the loss of internal structure, with the presence of multiple coalescing, nodular, well-defined and heterogeneous lesions with capsule deformity, appearing with an irregular profile. On dissection, the testicular parenchyma bulged over the cut section, confirming the increase in size. Microscopically, the lesion consisted mainly of large, densely packed, polygonal-to-round-shaped neoplastic cells. Mitotic figures were plentiful and frequently atypical; further microscopic features included apoptosis and necrosis. At immunohistochemistry, the entire neoplasm showed strong and diffuse immunolabeling for vimentin, while CD117-specific immunoreactivity was only observed in scattered clusters of neoplastic cells. Based on the gross, microscopic and IHC findings, a diagnosis of diffuse seminoma was made. Three months later, a follow-up examination showed no evidence of recurrence and the preservation of reproductive abilities. The case presented shows an unusual ultrasonographic pattern for seminoma and the basis of the correlation between the characteristics of the sonoelastographic examination and histological diagnosis.

## 1. Introduction

Seminomas are reported to be the most common testicular neoplasm in the equine stallion [1,2], as well as in humans [3] and dogs [4]. These arise from germ cells of the spermatic epithelium [5].

This type of germinal tumor affects horses of 11–22 years [6], but it is also reported in younger animals [7,8].

Although the literature includes many cases of seminoma affecting cryptorchid horses [9,10,11,12], the neoplasm can also affect descended testes [7,8,13,14,15]; in human medicine, cryptorchidism is found to increase chances of developing seminoma [3]. Indeed, a correlation between cryptorchidism and the onset of neoplasia has not been proven for horses [5].

Seminomas can be single, multiple, unilateral, bilateral, or cystic [7,11], but bilateral cases of seminomas are rare [1,9] and most are unilateral [10,11].

In contrast to other species, equine seminomas are frequently malignant, metastasizing to the abdominal organs, including the liver [9,16] and kidney [12], lymph nodes [9], peritoneum, mesentery and ureter, and also to the mediastinum [9,11,13]; nevertheless, clinical signs are mild and frequently nonspecific, such as weight loss [12] and nonsymmetrical testicular size [15].

Ultrasonography (US) is often used as a diagnostic tool in cases of testicular neoplasia, whose appearance ranges from circumscribed small nodules to large, complex masses with disruption of normal testicular anatomy; of note, different tumors cannot be distinguished based on US findings [11,13,14,15].

Next to B-mode findings, it is also possible to obtain information regarding the structural tissue abnormalities in a non-invasive way through a US-based technique called sonoelastography. This technique, indeed, is able to detect different levels of tissue elasticity based on the underlying disease that causes tissues abnormalities.

US elastography includes strain elastography (SE) and 2D shear-wave elastography (2D-SWE).

With SE, tissue elasticity is measured under the strain generated by external compression using an ultrasound transducer or internal physiologic pulsation such as heart beats or respiration; the acquired tissue elasticity data are encoded as color-mapping (typically red is used for soft tissues and blue for hard tissues) superimposed on the B-mode images and show the related elasticity of the anatomic lesion, with the possibility to compare the target tissue with a reference one through a semiquantitative analysis [17].

Two-dimensional SWE induces a shear wave in multifocal zones and measures the shear waves (SWs) that propagate at several locations around the acoustic pulse through a detection pulse. Subsequently, SWs inside the field of view (FOV) are integrated, and the distribution of SWs is displayed through a color map showing areas of high stiffness as red and low stiffness as blue.

The average SW is also quantified as Young’s modulus in kilopascals (kPa) and shear wave velocity measured in m/s by placing the ROI within the FOV [18].

In human medicine, sonoelastography, a non-invasive ultrasound-based technique to assess tissue stiffness, is highly sensitive to detect the difference in stiffness in healthy versus diseased testicles and has shown some promise in veterinary medicine; in particular, different levels have been observed of stiffness in the case of inflammatory and neoplastic disorders, with the highest stiffness observed in cases of malignant lesions [19,20]. This technique has also been proved to be useful to be applied to measure kidney stiffness in cats in the case of chronic kidney disease or to the detect canine liver fibrosis in a non-invasive way; furthermore, it is also potentially appliable in cases of musculoskeletal disorders [21,22,23,24].

In this work, we present the clinical case of an unusual unilateral seminoma in a Salernitano stallion, with a specific focus of the B-mode features of this neoplasia followed by the elastographic findings, with the aim to introduce the application of this technique in cases of reproductive disorders in stallions in order to potentially reduce, in the future, the need for invasive investigations to obtain the final diagnosis.

## 2. Materials and Methods

An 18-year-old Salernitano stallion, an endangered horse breed approved for reproduction, was referred to the Veterinary Hospital for a progressive enlargement of the left testicle, in the past eight months, with no other clinical signs.

Reproductive anamnesis reported normal sexual behavior and fertility, confirmed by a spermiogram performed during the previous reproductive season.

A general physical examination was performed, along with a focused examination of external genitalia, and basic hematological and serum biochemical tests, as a routine screening prior to any sedation or anesthesia. Trans rectal palpation and ultrasonographic examination (Logiq V2 sonographic system, GE Healthcare, Milan, Italy, and 7.5 mHz linear probe, GE Healthcare, Milan, Italy) were also performed, along with trans abdominal ultrasound examination (5 mHz convex probe). A mild sedation with 0.05 mg/kg of acepromazine (Prequillan, Fatro S.p.A., Ozzano dell’Emilia, Italy) was necessary to allow the procedures to be carried out safely.

B-mode ultrasound examination was performed with a Logiq S8 sonographic system (GE Healthcare, Milan, Italy) and 3–5 mHz convex probe (C1-5, GE Healthcare, Milan, Italy) on both testicles, in order to assess size, echogenicity, echotexture and borders; vascularization was assessed with color Doppler and B-Flow software with the same sonographic system.

Sonoelastography was performed with an 8.5–10 mHz linear probe using SE and 2D SWE software in the same sonographic system (GE Healthcare, Milan, Italy); in particular, since it was not possible to include the whole testicle in a single image field of view within the elastographic examination with the linear probe, each testicle was scanned including as much parenchyma as possible in the longitudinal plane and in the transverse plane, dividing the parenchyma into cranial, medial and caudal sections for the transverse plane.

Both SE and 2D-SWE were conducted by adapting the existing scanning protocol used for humans [25,26].

SE was conducted as follows: each section was scanned in order to acquire a 4–5 s loop, from which several frames representing the positive or negative peaks of the strain graph with the color of elastography images unchanged were collected, with the color bar indicative of the image quality appearing as green.

Qualitative evaluation of tissue stiffness was performed using a color scale adapted from the current human literature, in which normal parenchyma appears with a central parenchymal green color with blue edges surrounded by red bands. Testicular lesions, characterized by parenchymal abnormalities visible with B-mode US and resulting in a different elastogram appearance than the normal tissue, were classified following the classification explained in Table 1.

This scale is based on the basic elastography principle that the more rigid the tissue is, the higher the rate of malignancy is present, based on the positive correlation observed between stiff lesions and the rate of malignancy diagnosed with histopathology [20].

Two-dimensional SWE was then conducted, acquiring a cineloop of 20 s on each section, and then 5 elastograms were selected for each part, and three regions of interest (ROIs) of 5 mm were placed on the elastograms without overlapping on each to collect measurements in m/s [26].

Since specific elastography reference parameters are available for the equine species, the normality was assessed based on what was evidenced from the healthy testis.

Unilateral orchiectomy was performed under general anesthesia and allowed the complete removal of the affected testicle according the procedure described as follows.

The patient was prepared for a dorsally recumbent, open inguinal orchidectomy, under general anesthesia. Therefore, the patient was premedicated with medetomidine (7 μg/kg i.v.), and general anesthesia was obtained with an injection of ketamine (2.2 mg/kg i.v.) and diazepam (0.02 mg/kg i.v.) and maintained with isoflurane in oxygen-enriched air and CRI of medetomidine (3.5 μg/kg i.v.).

The surgical approach was inguinal. Skin was incised conveniently to identify the vaginalis comunis, which was sharply opened to grasp and withdraw the testis. The ligamentum caudae epididimi was divided, and the funiculus spermaticus sealed and divided with the LigaSure TM system [27].

Then, the testis was removed and the divided tissues surgically reconstructed lege artis.

Since no signs of regional lymphadenopathy were detected either clinically or sonographically, lymphadenectomy was not performed. To determine whether the testicular neoplasm had altered the normal hormonal profile of the stallion, blood samples for the evaluation of testosterone and estradiol serum concentrations were collected before the surgery (T0) and repeated three (T1) and five days (T2) after the removal of the pathological testis. Hormone detection was performed by chemiluminescence.

To determine the effects of the unilateral orchiectomy on stallion fertility, three months after surgery, during a follow up examination, a spermiogram was performed and compared with spermiograms obtained in the previous breeding season.

Tissue samples, representative of several areas of the testicle, were promptly fixed in 10% neutral buffered formalin, embedded in paraffin and routinely processed for histopathological investigations (hematoxylin and eosin stain). Moreover, tissue sections were tested by immunohistochemistry (IHC) for vimentin (murine monoclonal antibody, clone V9, Dako, final dilution = 1:100), cytokeratins (murine monoclonal antibody, clone AE1/AE3, Dako, Final dilution = 1:100) and CD117 (rabbit polyclonal antibody, Dako, dìfinal dilution = 1:100). To this aim, tissue sections were pre-treated with H2O2 (0.5% in methanol) for 20 min to quench endogenous peroxidase activity. Antigen retrieval was performed by heat treatment, at 96 °C for 45 min, in 0.01 M citrate buffer (pH 6). Primary antibodies were incubated overnight at 4 °C, and the immune reactions were visualized by means of a streptavidin peroxidase technique (Dako RealTM Detection System) using 3,3′ diaminobenzidine tetrahydrochloride as a chromogen. Suitable positive controls were included in each run, and negative controls were obtained by omitting the primary antibodies.

As far as it regards the data obtained from elastography, statistical analysis was performed on the data expressed in m/s with MATLAB R2015a software; in particular, the distribution of data was analyzed using the Shapiro–Wilk test, and the comparison between the two testicles was performed with Wilcoxon signed-rank test with statistical significance defined as an alpha level of *p* < 0.05.

## 3. Results

Physical examination was normal for the species and breed (heart rate 44 bpm; respiration 12 rpm; temperature 37.3 °C).

Clinical evaluation, trans rectal palpation and trans rectal and abdominal ultrasonography did not show any alteration in the explorable lymph nodes and organs; the size of the left enlarged testicle, detected with the caliper, was 15.8 cm CrCd × 8.7 cm DV × 11 cm LL (Figure 1); the right testis measured 11.7 cm CrCd × 5.1 cm DV × 8.3 cm LL, including the epididymal tail.

Palpation of the testis showed a considerable parenchyma inhomogeneity, losing its normal turgid resilient consistency. In fact, the testicle appeared to consist of harder areas and areas of decreased consistency, separated by harder septa, of a fibrous-like consistency. The evaluation did not cause pain, and the normal mobility of the testis with no scrotal adhesions was assessed.

A complete physical examination showed no evidence of metastases.

Hematological and serum biochemical tests showed no alterations.

Comparison of spermiogram results is shown in Table 2.

The ultrasound examination showed that the right testicle appeared normal (Figure 2), while the left testicle severely enlarged (13.35 cm CrCd × 8.19 cm DV × 9.42 cm LL vs. 9.25 cm CrCd × 4.75 cm DV × 6.91 cm LL). The parenchyma was characterized by the presence of multiple coalescing, nodular, well-defined and heterogeneous lesions of various size that almost completely effaced the normal anatomy and deformed the capsule or distorted normal shape (Figure 3); color Doppler and B-Flow evidenced a diffuse increase in vascularization, characterized by vessels of different size and localized both peripherally and in the central portion of the various nodular lesions, instead of normal unform vessel distribution from the capsule to the testicle mediastinum (Figure 4).

Regarding SE results, since it was technically not possible to include the whole testicle in one single examination, the color-coded classification was performed considering the most prevalent pattern within the image.

The right testicle was characterized using central parenchymal red-green color with blue edges and without red bands (Figure 5A,B), while the left one was characterized by a heterogeneous appearance, predominantly classifiable as 5, mostly in the central areas, with some areas with 1 and 2 scores localized more peripherally (Figure 5C,D).

Regarding 2D-SWE, the left testicle was globally significantly stiffer (*p* < 0.0028) than the contralateral, with a median value for the left testicle of 3.66 m/s (1.8–6.64), and median values for the right testicle were 1.15 m/s (0.8–1.6) (Figure 6A–D).

Based on the ultrasonographic findings, a neoplasm was suspected.

Surgery was performed without complications, and all the pathological testicular tissue was removed by surgical ligation on the spermatic cord that appeared normal, reducing the possibility of the dissemination of neoplastic cells.

The hormonal profile showed a significant reduction in testosterone levels after surgical removal, showing values of about half, while estradiol assays showed only a slight reduction. Values are shown in Table 2.

At dissection, the parenchyma bulged markedly when the testis was incised, thus confirming its increase in size. Macroscopically, the testicular tissue was completely disrupted, with the normal appearance of the parenchyma being effaced (Figure 7).

Microscopically, the lesion mainly consisted of large, densely packed, polygonal-to-round shaped neoplastic cells. In some areas, such cells were organized as cords, small clusters or tubular-like structures, surrounded by loose connective tissue. Anisocytosis and anisokaryosis were always prominent. Most of neoplastic cells had a single nucleus, although bi- or multinucleate cells were often detected. Mitotic figures were plentiful and frequently atypical (1–3 mitoses/hpf). Lymphocytes were numerous, and frequently organized as large aggregates. Apoptosis and necrosis represented further microscopical features.

At IHC, neoplastic cells proved to be negative for cytokeratins, but they showed a strong cytoplasmic immunoreactivity for vimentin. Clusters of cells were also positive for CD117, with a cytoplasmic and/or a membranous pattern of immunoreactivity.

Considering gross, microscopic and IHC findings, a definitive diagnosis of diffuse seminoma was made (Figure 8).

Three months after surgery, a follow-up evaluation showed no other changes in the right testicle. Hematology and serum-biochemical panels were repeated, with normal results. A complete spermiogram was performed to assess fertility, showing results comparable to those obtained prior to the onset of neoplasm. A comparison of spermiogram results is shown in Table 3. In the breeding season following the unilateral orchiectomy, the stallion was used in natural service, providing a pregnancy rate of about 80%. Periodic medical examinations are scheduled to check for metastases.

## 4. Discussion

Seminomas are the most common testicular neoplasms in horses [2], although tumors of the male gonads are considered uncommon in this species [1] because most horses are castrated at an early age [5].

The age of appearance of seminoma can range from 7 [7,8] to 22 years [6], and both retained [12] and descended testicles can be affected [8]. More frequently, this gonadal tumor is reported to be unilateral [10], although bilateral cases have been described [9].

Equine seminoma can show a malignant course [11], but clinical signs are usually mild and nonspecific [12,15].

This report describes an 18-year-old stallion, presenting a progressive enlargement of the left descended testis, with no other clinical signs. Such a nonspecific clinical presentation is clearly not suggestive of seminoma, despite the consulted literature reporting cases of horses affected by seminomas, in the absence of any clinical signs, except weight loss [7]. Additionally, the age of occurrence matches reported data [9,11,12,14].

The clinical finding of an enlarged non-painful testis should be differentiated from other non-painful lesions, such as hydrocele and hematocele [15], or other testicular neoplasms, such as sertolioma or leydigoma [2]. On the other hand, some papers report a preventive hemicastration in the absence of a confirmed diagnosis, to allow gross and microscopic analysis after the removal of the mass [7,8,14].

Little information is available about the US features of equine seminoma, which has been described as a mass with variable echogenicity or diffusely hypoechoic with ill-defined regions of hyperechogenicity [12].

In our clinical case, compared to what has been previously described, the affected parenchyma appeared markedly nodular, rather than characterized by ill-defined heterogeneous areas.

For sonoelastography, both SE and 2D-SWE evidenced a difference in stiffness between the two testicles, with both qualitative and quantitative evaluations, since with SE, a different color distribution was evident comparing the diseased and the healthy one, with a similar trend quantitatively expressed by the measurements in m/s.

The data described herein are consistent with what is currently observed in human medicine, where elastography is highly sensitive for use in non-invasively detecting focal lesions in the testicular parenchyma [20], especially in the case of neoplastic disorders; furthermore, in the dog, different elastographic software was able to assess differences in stiffness in cases of diseases of varying etiologies [19].

The main difference and limitation of the elastography application (both SE and 2D-SWE) in the case described herein is the impossibility of including all testicular parenchyma in one examination.

This might have partially influenced the final results, especially for the variable SE color pattern in the neoplastic tumor, since this technique is based on the manual compression exercised by the operator; the different scan planes, combined with having scanned different sections of the parenchyma might have caused a difference in the mechanical pressure distribution, which might explain the non-homogeneous color pattern in different testicle sections.

This variability could also be due to the presence of necrosis next to the neoplastic tissue, thus causing a heterogeneous color pattern indicating different grades of stiffness within the same image.

Particularly for SE, the choice to consider only the qualitative evaluation on the elastogram, instead of also including the semiquantitative analysis, was made because in human medicine, SE is typically performed by comparing the focal lesion with the background normal testicle at same depth [28], and in our case, it was not possible to perform such a comparison because it was not possible to include sonographically healthy and diseased parenchyma in the same image with the linear probe.

Furthermore, in this case, a structural difference between the two testicles was relatively obvious from the clinical and B-Mode examinations; introducing sonoelastography into routine examinations could be useful to identify lesions potentially able to impair the reproductive condition of the stallions before the appearance of clinical signs.

Overall, pathological findings observed herein (i.e., the lobulated appearance of the neoplasm at gross inspection, the microscopic features of neoplastic cells, their immunoreactivity for CD117 and the marked infiltration of lymphocytes) match with those reported in seminomas in the horse and in other animal species.

Regarding the hormonal profiles, results show a marked decrease in serum testosterone concentrations and a mild decrease in blood or serum estrogen in the days after surgery. Recent research by Valdez and colleagues [29] also shows a decrease in testosterone immediately after unilateral orchiectomy, but he also reports an increase in this steroid 90 days after surgery. A similar trend has been demonstrated for serum estrogen concentrations [29]. We did not detect or repeat a long-term hormonal profile, but a trend similar to the one found in the cited literature can be assumed, considering the full recovery of the seminal parameters, three months after surgery, and of the normal sexual behavior.

These findings are justified by the bibliography consulted, as it is well-known that hemicastration determines an increase in the serum Follicle Stimulating Hormone (FSH) [30] and Luteinizing Hormone (LH) [31], associated with a compensatory hypertrophy of the remaining testicle [29,30,31].

The above-cited physical and hormonal modification, following the hemi-orchidectomy, determine the complete recovery of a normal fertility, as also demonstrated in this report comparing spermiograms pre- and post-surgery. The positive outcome after unilateral castrations is supported by consulted references [10,32,33].

B-mode ultrasonography is part of the clinical routine for evaluating the stallion’s fertility [34]; however, little is known about ultrasonographic features of equine seminoma [12]. Although a certain diagnosis can only be obtained with biopsy or histology after the removal of the neoplastic testis [34,35], this article highlighted the sensitivity of 2D-SWE and SE in detecting changes in the testicular parenchyma and its potential for use in routine clinical practice as a non-invasive diagnostic technique for stallion testicular tumors.

## Figures and Tables

**Figure 1 animals-12-00936-f001:**
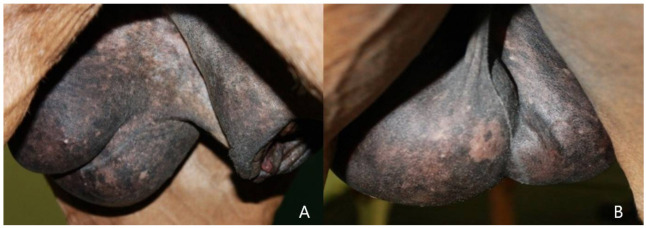
Medio-lateral (**A**) and caudo-cranial (**B**) view of the scrotum, showing an increase in the volume of the pathological testis (left) compared to the contralateral (right).

**Figure 2 animals-12-00936-f002:**
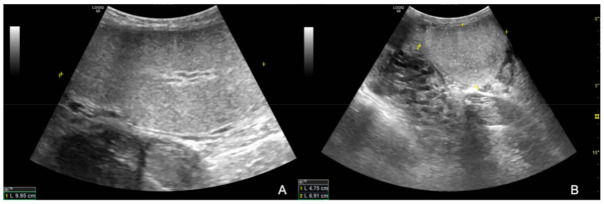
B-mode longitudinal (**A**) and transverse (**B**) images of the right testicle; note the homogeneous appearance of normal testicular parenchyma.

**Figure 3 animals-12-00936-f003:**
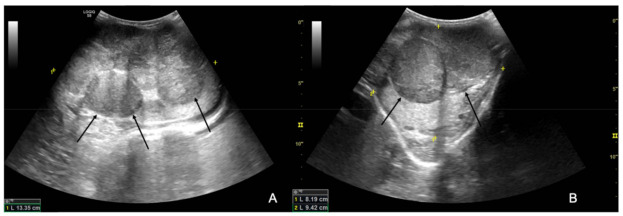
B-mode longitudinal (**A**) and transverse (**B**) images of the left testicle; the black arrows point at the heterogeneous and well-defined nodular lesions effacing the normal testicular parenchyma.

**Figure 4 animals-12-00936-f004:**
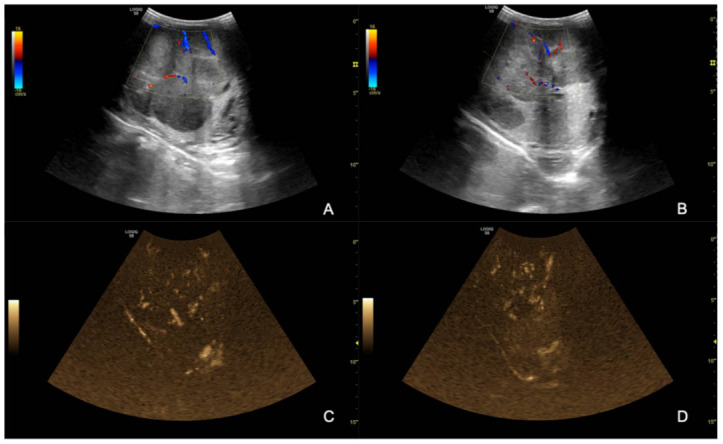
B-blow images (**A**,**B**) and Color Doppler (**C**,**D**) of the left testicle; note the heterogeneous vascular lesion distribution appearing as both peripheral and central in different portions of the same testicle.

**Figure 5 animals-12-00936-f005:**
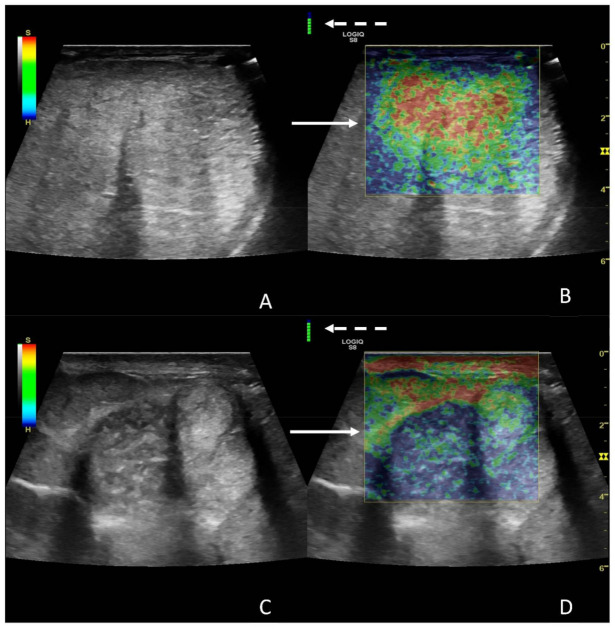
Images from SE of a normal (**A**,**B**) and a diseased testicle (**C**,**D**). In panel A and B, the B-mode image and the elastogram are presented, respectively; in the elastogram, notice the color distribution, characterized by central parenchymal red-green color with blue edges (white solid arrow). In panels C and D, B-mode image and elastogram are presented as well, with the elastogram characterized by an heterogenous pattern, mostly blue with a central small green areas (score 5—white solid arrow). The white dotted arrows point at a color bar is indicative of the quality of examination, which is considered good if it appears green.

**Figure 6 animals-12-00936-f006:**
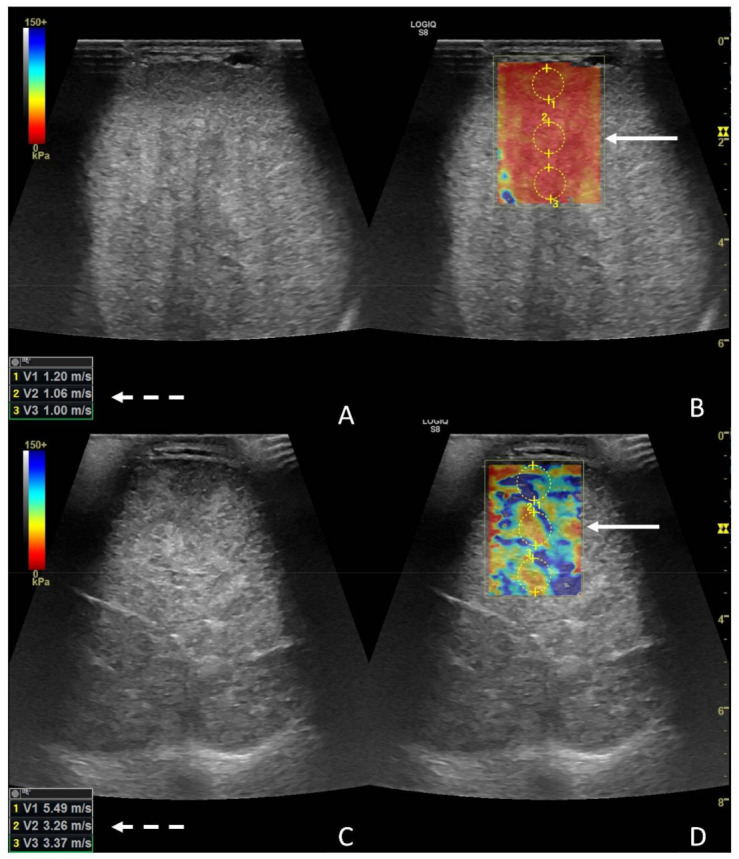
Images from 2D-SWE of a normal (**A**,**B**) and a diseased testicle (**C**,**D**). In panels A and B, the B-mode image and the elastogram are presented, respectively; the white dotted arrow points at the values from the ROIS manually drawn on the elastogram (white solid arrow). In panels C and D, a B-mode image and elastogram are presented as well; notice the higher values (white dotted arrow) from the ROIS pointed on the elastogram (white solid arrow).

**Figure 7 animals-12-00936-f007:**
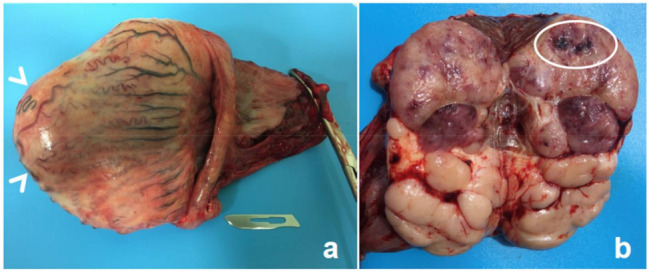
Horse testis. At the external inspection (**a**), the shape of the testicle is partially altered, due to the presence of a large bulge (white arrowheads). On the cut section (**b**), the features of the testicular parenchyma are completely modified. About 50% of the surface appears to be invaded by a cluster of whitish, lardaceous nodules (lower part of the picture), while the remaining part was reddish-to-greyish, with hemorrhagic foci also being evident (bounded with a white line).

**Figure 8 animals-12-00936-f008:**
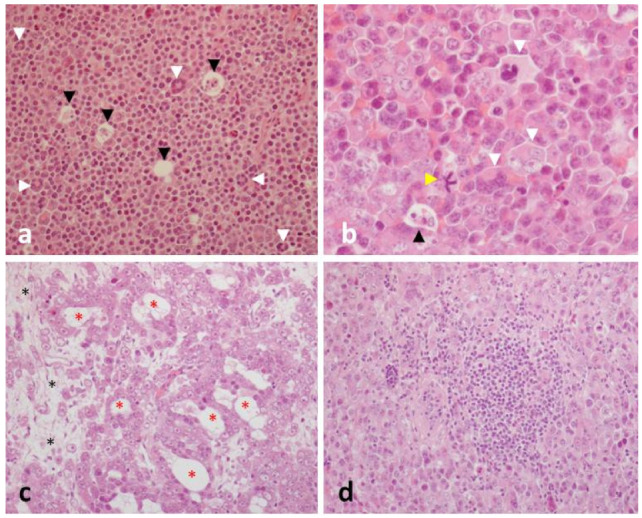
Horse testis. (**a**) Neoplastic cells are densely packed, and multinucleated cells can be easily detected (white arrowheads). Individual cell death (black arrowheads) determines a starry-sky pattern. (**b**) At a higher magnification, the main features of the neoplastic cells are better appreciated. Most of them are mononuclear and polygonal in shape, with a central, often vesicular nucleus, surrounded by a more or less abundant eosinophilic cytoplasm. Prominent nucleoli are also evident. Multinucleated cells usually have their nuclei arranged in a horseshoe-shaped pattern, at the cell periphery. The apoptosis of a single cell (black arrowhead) and an atypical mitosis (yellow arrowhead) are also seen. (**c**) Neoplastic cells are arranged as cords, tubular-like structures (red asterisks) embedded within nucleoli are very evident, and the stroma shows a loose appearance. (**d**) A lymphocytic aggregate is shown embedded within the neoplastic proliferation. Hematoxylin and eosin stain. Final magnification: ×200 (**a**,**c**,**d**), ×400 (**b**).

**Table 1 animals-12-00936-t001:** SE elastographic color coded scale classification.

Grading Scale	Color Coded Elastogram Appearance
1	The lesion is almost completely green but with some red spots.
2	The entire lesion is homogeneously green.
3	The lesion is almost completely green but with some small blue spots.
4	The lesion is green at the periphery with a central blue area.
5	The lesion is almost completely blue with central small green or red areas
6	The lesion is completely blue

**Table 2 animals-12-00936-t002:** Serum concentration of testosterone and estradiol, measured with chemiluminescence. T0, before Surgery; T1, 3 days after surgery; T2, 5 days after surgery.

	Testosterone (ng/mL)	Estradiol (pg/mL)
**T0**	6.91	292
**T1**	3.26	276
**T2**	3.12	253

**Table 3 animals-12-00936-t003:** Comparison of spermiograms performed at *t*0 (prior to tumor onset) and at *t*1 (three months after surgery). Volume was measured using a graduated container; concentration was assessed using a Burker Chamber; Total Motility (TM) and Progressive motility (PM) were evaluated using a CASA System, set to 30 frames per fields and 60 Hz per rate (CASA IVOS, Hamilton Torne). Vitality was assessed using a fluorescent stain with propidium iodine.

Parameters	*t*0	*t*1	Methods
Volume	**Total (mL)**	**Total (mL)**	Graduated container
	110	72	
	**Gel free (mL)**	**Gel free (mL)**	
	85	53	
Concentration	**(×10^6^ sperm/mL)**	**(×10^6^ sperm/mL)**	Burker chamber
	56	38	
Motility	**TM %**	**TM %**	CASA
	71	69	
	**PM %**	**PM %**	
	55	56	
	**Normal cells (%)**	**Normal cells (%)**	
Morphology	65	67	CASA
	**Abnormalities (%)**	**Abnormalities (%)**	
	35	33	
	**Other cell types**	**Other cell types**	
	Presence of particulate matter on the bottom	Presence of particulate matter on the bottom	
Vitality	**%**	**%**	Propidium Iodide
	55	57	

## Data Availability

Not applicable.

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
