# Peer review of "Ultrasound Examination of Unilateral Seminoma in a Salernitano Stallion"

_animals, 2022, doi:10.3390/ani12070936_

Round 1

Reviewer 1 Report

Dear Authors, please correct the marked sentences and figures!

Author Response

Authors do not understand what is wrong with the word “blue”.

The Fig. 8 has been corrected.

Reviewer 2 Report

The manuscript is generally well written and is describing an interesting case of a testicular tumor in a horse. Unfortunately, the novelty of the work is missing and in my opinion there is very little additional clinical or scientific merit which has not been provided by previous publications.

General comments

Abstract:

It would be beneficial to add the duration of clinical signs (enlargement) here and later on in the manuscript.

Line 20-21: This sentence reads as if you performed the diagnostic procedures after removal of the testis. Please revise accordingly.

Line 26/27: please add how many mitotic figures per hpf. Be in your description is detailed as possible even in the abstract.

Line 33/34: How is this shown. This should be clear in the results of the abstract already. Additionally, where is the novelty here? What is interesting for the reader? Why should someone read your case report?

Introduction:

Line 59-66: This section should be at the end of your introduction .

Line 70-87: These section would be ideally moved up and connected with the section in lines 66-69.

Generally, you need to state clearly what the benefits of your additional diagnostics are! Where is the novelty here and why should this additional step be taken if removal is in most cases curative and probably it would be more beneficial to try and visualise the adjacent lymphnodes since metastasis might be a potential limiting factor.

Sections 2 and 3 (M&M and Results): Generally, case reports contain a case summary and not materials and methods and results as it results in some duplication of descriptions but this is something that need to be discussed with the editor of the journal.

Lines 95-101/166-175: General physical examination parameters should be listed ( e.g. heart rate, RR, rectal temp). I would also like to know how the authors excluded metastasis based on physical examination since moth commonly this occurs in abdominal organs or lymphnodes.

Was the horse sedated for the ultrasonographic evaluation and if so please add drugs and doses according to journal guidelines.

Lines 136-141: It is not clear what data was assessed for normality and what measures did you exactly compare between the two testis? This needs to be clearly stated here.

Line 141-143/200-202: I would like to see a bit more detail on general anaesthesia protocol and castration technique. Did you check the adjacent lymphnodes and take biopsies to exclude metastasis? And if you did not, why?

Line 142-147: Was there a specific reason for the blood tests? It has been well documented that while hormonal levels drop shortly, generally fertility is not affected by unilateral castration in horses…

I am unfortunately feeling, you are proving a lot of points here that has already been well established and described, including US of the testis.

Lines 119-128: The classification should be placed in a table format.

Lines 187-195: This section reads very subjective and I am questioning repeatability of the obtained results. In order to establish a new technique you need to establish objective measurements that can be repeated and used in other cases consistently.

Discussion: While there is a good section on limitations I am missing an good argument why this diagnostic technique is necessary. I am not agreeing this your statement that your physical examination findings are not suggestive of this tumor. In fact, I believe it is a textbook case and while the specific tumor type might not always be apparent, it will not change the course of action which is removal of the diseased testes.

I am also not agreeing with your statement (lines 317-322) that the ultrasonographic description of the published case of a seminoma does sound similar enough to me to include this type of tumor in the differential diagnosis list.

Author Response

REV 2

The manuscript is generally well written and is describing an interesting case of a testicular tumor in a horse. Unfortunately, the novelty of the work is missing and in my opinion there is very little additional clinical or scientific merit which has not been provided by previous publications.

 General comments

Abstract:

It would be beneficial to add the duration of clinical signs (enlargement) here and later on in the manuscript.

 It has been added, here and in line 92, that clinical signs appeared in the last 8 months prior to the evaluation.

Line 20-21: This sentence reads as if you performed the diagnostic procedures after removal of the testis. Please revise accordingly.

 Text has been corrected as follow: “An ultrasound evaluation was performed, along with hormonal profile. A histopathological evaluation of the testis was performed after unilateral orchiectomy

Line 26/27: please add how many mitotic figures per hpf. Be in your description is detailed as possible even in the abstract.

Corrected accordingly.

Line 33/34: How is this shown. This should be clear in the results of the abstract already. Additionally, where is the novelty here? What is interesting for the reader? Why should someone read your case report?

The novelty of the data herein reported relies in the description of both B-mode and sonoelastography features of unilateral seminoma in a Salernitano stallion. As it has been described in the text, seminomas have been described as a mass with variable echogenicity or diffusely hypoechoic with ill-defined regions of hyperechogenicity, while in our report the US examination revealed a strongly nodular appearance of such neoplasia; furthermore, next to the strictly B-mode findings, we present the data related to the elastographic examination of both testes, in order to evidence that this technique, next to the all the possible applications, could be applied also in case of stallion reproductive disorders. The text has been modified accordingly.

Introduction:

Line 59-66: This section should be at the end of your introduction .

The text has been modified accordingly. 

Line 70-87: These section would be ideally moved up and connected with the section in lines 66-69.

This section has been modified including what has been suggested in the previous comments.

Generally, you need to state clearly what the benefits of your additional diagnostics are! Where is the novelty here and why should this additional step be taken if removal is in most cases curative and probably it would be more beneficial to try and visualise the adjacent lymphnodes since metastasis might be a potential limiting factor.

The benefits of additional diagnostics relies in the possibility to potentially distinguish different disorders in a non-invasive way without the need of further tissue biopsies, as it is currently performed in human medicine. Furthermore, also in veterinary medicine, sonoelastography has been proved to be useful to distinguish different testicular disorders (see Feliciano et al, 2016). Based on these data, we proposed an elastographic protocol in this specific clinical case, with the aim to verify if there was a difference between the healthy and the neoplastic testis; our results evidenced that elastography evidenced a substantial difference in tissue stiffness, thus making this technique appliable to other testicular disorders also in the equine species.

Sections 2 and 3 (M&M and Results): Generally, case reports contain a case summary and not materials and methods and results as it results in some duplication of descriptions but this is something that need to be discussed with the editor of the journal.

Since there is no agreement on this point between the reviewers, we leave the article in this format and will be the editor to suggest the proper changes. 

Lines 95-101/166-175: General physical examination parameters should be listed ( e.g. heart rate, RR, rectal temp). I would also like to know how the authors excluded metastasis based on physical examination since moth commonly this occurs in abdominal organs or lymphnodes.

Clinical findings have been added in the “results” section (specify lines).

In lines 98-100 we added the following statement: “Trans rectal palpation and ultrasonographic examination (Logiq S8 sonographic system, GE Healthcare, Milan, Italy, and 7.5 mHz linear probe, GE Healthcare, Milan, Italy) were also performed, along with trans abdominal ultrasound examination (5 mHz convex probe).”, as a complete clinical and andrological evaluation must include trans rectal palpation and ultrasonography. These procedures allow clinicians to detect any anatomical changes of the organs that can be explored.

In lines 186-189 the statement has been modified as follows “Physical examination was normal for the species and breed (heart rate 44 bpm; respiration 12 rpm; temperature 37,3°C). Clinical evaluation ruled out any involvement of the draining lymph nodes; Trans rectal palpation and trans rectal and abdominal ultrasonography did not show any alteration of the explorable organs”.

These findings allowed authors to exclude metastasis. Anyhow, the stallion is subjected to regular checks, as stated in lines 253-254, to check for metastasis.

Was the horse sedated for the ultrasonographic evaluation and if so please add drugs and doses according to journal guidelines.

the statement “A mild sedation with of acepromazine (0.05 mg/kg) was necessary to allow the procedures to be carried out safely” has been added in lines 101-102

Lines 136-141: It is not clear what data was assessed for normality and what measures did you exactly compare between the two testis? This needs to be clearly stated here.

Since specific elastography reference parameters are available for the equine species, the normality was assessed based on what was evidenced from the healthy testis. This statement has been added in the text. 

Line 141-143/200-202: I would like to see a bit more detail on general anaesthesia protocol and castration technique. Did you check the adjacent lymphnodes and take biopsies to exclude metastasis? And if you did not, why?

Specific details regarding the surgery have been added in the text; Since no sign of regional lymphadenopathy were detected both clinically and sonographically, lymphadenectomy was not performed.

 The statement “The patient was prepared for a dorsally recumbent, open inguinal orchidectomy, under general anaesthesia. Therefore was premedicated with medetomidine (7 μg/kg i.v.) and general anaesthesia was obtained with an injection of ketamine (2.2 mg/kg i.v.) and diazepam (0.02 mg/kg i.v.) and maintained with isoflurane in oxygen-enriched air and CRI of medetomidine (3.5 μg/kg i.v.). The surgical approach was inguinal. Skin was incised conveniently to identify the vaginalis comunis, that was sharply opened to grasp and withdraw the testis. The ligamentum caudae epididimi was divided and the funiculus spermaticus sealed and divided with the LigaSure TM system. Then the testis was removed and the divided tissues surgically reconstructed lege artis.” has been added, as requested.

Line 142-147: Was there a specific reason for the blood tests? It has been well documented that while hormonal levels drop shortly, generally fertility is not affected by unilateral castration in horses…

The statement has been modified as follows: “To determine whether the testicular neoplasm had altered the normal hormonal profile of the stallion, blood samples for the evaluation of testosterone and estradiol serum concentrations were collected before the surgery (T0) and repeated three (T1) and five days (T2) after the removal of the pathological testis. Hormone detection was performed by chemiluminescence.”.

I am unfortunately feeling, you are proving a lot of points here that has already been well established and described, including US of the testis.

 Hormonal assays have been carried out as a part of a complete andrological evaluation as it’s known that other testicular neoplasm can produce androgens and estradiol (Schumacher et al, 2009).

Data obtained agree with the literature available on equine seminomas and rule out other neoplasm as sertolioma and leydigoma.

The novelty of this report is not clinical, histological or laboratory diagnosis of seminoma, but the unusual ultrasonographic pattern and the use of sonoelastography. In this context, any other finding is useful to support the correlation between US findings and seminoma.

Lines 119-128: The classification should be placed in a table format.

The text has been modified accordingly.

Lines 187-195: This section reads very subjective and I am questioning repeatability of the obtained results. In order to establish a new technique you need to establish objective measurements that can be repeated and used in other cases consistently.

The procedure herein described is basically standardized based on the present literature in both human veterinary medicine in terms of scanning protocol, color coded elastogram interpretation and number of measurements that have been clearly pointed in the text. The only potentially subjective aspect of this protocols relies in the proper US window to perform both US and elastogram, but, in the author opinion, this aspect depends on the underlying disorders and also the size of the structure analyzed; further studies on a larger scale are needed to assess if there could be biases in case of different sizes of the analyzed structures.

Discussion: While there is a good section on limitations I am missing an good argument why this diagnostic technique is necessary. I am not agreeing this your statement that your physical examination findings are not suggestive of this tumor. In fact, I believe it is a textbook case and while the specific tumor type might not always be apparent, it will not change the course of action which is removal of the diseased testes.

Authors are aware that a tumour can be easily suspected from the clinical examination, but, in the authors opinion, to fully characterize the underlying disease, additional imaging examinations (e.g US and/or sonoelastography) could provide valuable information and complete/confirm the clinical suspect in order to fully determine the subject prognosis.

I am also not agreeing with your statement (lines 317-322) that the ultrasonographic description of the published case of a seminoma does sound similar enough to me to include this type of tumor in the differential diagnosis list.

The authors agree with the reviewer and in the description of the case a neoplasia was suspected, without considering a seminoma as first differential diagnosis based on the nodular aspect that has not been mentioned by previous published data; that’s the reason why the authors considered interesting and useful to report these findings.

Reviewer 3 Report

The manuscript “Ultrasound examination of unilateral seminoma in a Salernitano stallion” aims to describe a clinical case of a unilateral seminoma, reporting the sonographic and sonoelastographic appearance of the neoplasm. The article is interesting, but it should be well described. The materials and methods should be better organized as well as the results. For this reasons the manuscript can be  suitable to publish, after revision. Following specific comments.

Line 16-17: change type of characters

Line 52: delete “liver” that is repeated

Line 55: add (US)

Line 59: better describe the aims of the report

Material and methods: divide the section in subparagraphs

Line 102: 9L?

Line 103: delete brackets

Line 103: add software company

Line 109: add reference

Line 134: add Region of Interest (ROI)

Line 144-147: better specify analysis

Line 137-141: move this sentence at the final part of MM section

Line 150: specify that spermiogram was performed before the surgery and how it was performed

Results: review the number of the table

Line 172: how did you value fibrous septae, at palpation?

Line 183: delete comma

Line 187: figure 3 was in B-mode

Line 254-259: it belongs to the legend of the figure. 

Author Response

REV 3

The manuscript “Ultrasound examination of unilateral seminoma in a Salernitano stallion” aims to describe a clinical case of a unilateral seminoma, reporting the sonographic and sonoelastographic appearance of the neoplasm. The article is interesting, but it should be well described. The materials and methods should be better organized as well as the results. For this reasons the manuscript can be  suitable to publish, after revision. Following specific comments.

Line 16-17: change type of characters

Correction done

Line 52: delete “liver” that is repeated

Correction done

Line 55: add (US)

Correction done

Line 59: better describe the aims of the report

Aims have been properly updated

Material and methods: divide the section in subparagraphs

Since there is no agreement on this point between the reviewers, we leave the article in this format and will be the editor to suggest the proper changes.

Line 102: 9L?

The probe details have been modified

Line 103: delete brackets

Line 103: add software company

This information has been added

Line 109: add reference

References have been added in the text.

Line 134: add Region of Interest (ROI)

Text has been properly modified

Line 144-147: better specify analysis

The statement has been modified as follows: “To determine whether the testicular neoplasm had altered the normal hormonal profile of the stallion, blood samples for the evaluation of testosterone and estradiol serum concentrations were collected before the surgery (T0) and repeated three (T1) and five days (T2) after the removal of the pathological testis. Hormone detection was performed by chemiluminescence.”.

Line 137-141: move this sentence at the final part of MM section

Text has been updated accordingly

Line 150: specify that spermiogram was performed before the surgery and how it was performed

The statement was modified as follows: “a spermiogram was performed and compared with spermiograms obtained in the previous breeding season”. (as stated in line 95).

the description of the spermiogram method is described in the caption of Table 2.

Results: review the number of the table

Tables have been accordingly reviewed

Line 172: how did you value fibrous septae, at palpation?

Text corrected

Line 183: delete comma

Correction done.

Line 187: figure 3 was in B-mode

Text has been properly modified.

Line 254-259: it belongs to the legend of the figure. 

Text has been modified accordingly.

Round 2

Reviewer 2 Report

Please see comments below